A quick and robust method for quantification of the hypersensitive response in plants

Johansson Oskar N.
Nilsson Anders K.
Gustavsson Mikael B.
Backhaus Thomas
Andersson Mats X.
Ellerström Mats mats.ellerstrom@gmail.com
Department of Biology and Environmental Sciences, University of Gothenburg , Gothenburg , Sweden
Renaut Jenny
Electronic publication date: 2015 Dec 1
Publication date: 2015
Volume: 3
Electronic Location ID: e1469
Received 2015 Aug 25; Accepted 2015 Nov 12
Copyright: © 2015 Johansson et al.
Copyright year: 2015
Copyright holder: Johansson et al.
License: This is an open access article distributed under the terms of the Creative Commons Attribution License, which permits unrestricted use, distribution, reproduction and adaptation in any medium and for any purpose provided that it is properly attributed. For attribution, the original author(s), title, publication source (PeerJ) and either DOI or URL of the article must be cited.
License URL: https://creativecommons.org/licenses/by/4.0/

Keywords: Hypersensitive response (HR), Programmed cell death (PCD), Electrolyte leakage, Effector-triggered-immunity (ETI), Pseudomonas syringae, AvrRpm1

Funding: Swedish Council for Environment, Agricultural Sciences and Spatial Planning 2007-1051 2007-1563 2009-888 Olle Engkvist Byggmästare, Adlerbertska research foundation The financial support of the Swedish Council for Environment, Agricultural Sciences and Spatial Planning to Mats Ellerström (project No. 2007-1051) and Mats Andersson (project No. 2007-1563 and 2009-888), the Olle Engkvist Byggmästare, Adlerbertska research foundation to Mats Andersson is gratefully acknowledged. The funders had no role in study design, data collection and analysis, decision to publish, or preparation of the manuscript.

==============================
One of the most studied defense reactions of plants against microbial pathogens is the hypersensitive response (HR). The HR is a complex multicellular process that involves programmed cell death at the site of infection. A standard method to quantify plant defense and the HR is to measure the release of cellular electrolytes into water after infiltration with pathogenic bacteria. In this type of experiment, the bacteria are typically delivered into the plant tissue through syringe infiltration. Here we report the development of a vacuum infiltration protocol that allows multiple plant lines to be infiltrated simultaneously and assayed for defense responses. Vacuum infiltration did not induce more wounding response in Arabidopsis leaf tissue than syringe inoculation, whereas throughput and reproducibility were improved. The method was used to study HR-induced electrolyte loss after treatment with the bacterium Pseudomonas syringae pv. tomato DC3000 harboring the effector AvrRpm1, AvrRpt2 or AvrRps4. Specifically, the influence of bacterial titer on AvrRpm1-induced HR was investigated. Not only the amplitude, but also the timing of the maximum rate of the HR reaction was found to be dose-dependent. Finally, using vacuum infiltration, we were able quantify induction of phospholipase D activity after AvrRpm1 recognition in leaves labeled with 33PO4.

Introduction

Phytopathogenic microorganisms are omnipresent in nature and pose a constant threat to plants. Despite this, plants rarely become infected and develop disease; a multi-layered innate immune system protects them from most pathogens. Plants can perceive microbes through pattern recognition receptors that recognizes conserved microbe-associated molecular patterns (MAMP), leading to activation of defense mechanisms and MAMP-triggered immunity (MTI) (Dodds & Rathjen, 2010). Consequently, pathogens have evolved several strategies which enable them to break plant defense and proliferate. The pathogens may for example secrete effector molecules that suppress MTI (Arnold & Jackson, 2011) or promote an outflow of nutrients from the host (Chen et al., 2010). Plants have in turn acquired immune receptors called resistance (R) proteins that recognize pathogenic effectors directly or via proteome perturbations and activate defense pathways (Spoel & Dong, 2012). The outcomes of effector-triggered-immunity (ETI) include an oxidative burst, production of anti-microbial compounds, expression of pathogenesis-related (PR) genes, and often a rapid programmed cell death (PCD) at the site of infection. The reaction as a whole is denoted hypersensitive response (HR) (Coll, Epple & Dangl, 2011; Hofius et al., 2011; Mur et al., 2008). All aspects of the HR are not prerequisites for resistance, as exemplified by several Arabidopsis thaliana (hereafter Arabidopsis) mutants showing reduced pathogen triggered PCD but unaffected in their ability to restrict growth of avirulent bacteria (Johansson et al., 2014; Jurkowski et al., 2004; Yu, Parker & Bent, 1998). HR induced by avirulent bacteria depends on living cells of the pathogen and can be activated in the plant within 15 min after recognition of the intruder (Klement & Goodman, 1967).

In the post-genomic era, enormous progress has been made in the understanding of the underlying genetics of plant immunity. Large scale transcriptomic and proteomic studies have identified thousands of genes and gene products as directly or indirectly responding to effector elicitation (Katagiri, 2004; Zimaro et al., 2011). Candidate genes from such studies may later be tested for their role in plant immunity by the use reverse genetics. This has evoked the need for quick and effective screening methods for the identification of immune-depressed or hyperactive immune mutants.

Various read-outs and methods have been employed to quantify plant defense responses after pathogen elicitation. For example, the area of cell death (Alamillo & Garcia-Olmedo, 2001), K+/H+ fluxes (Atkinson, Huang & Knopp, 1985), abundance of PR proteins (Sels et al., 2008), and DAB staining for reactive oxygen species (Thordal-Christensen et al., 1997) have all been used as read-outs for quantifying HR. The PCD of the HR is intimately associated with loss of electrolytes from the dying cells. Hence, the extent of the HR related PCD (HR-PCD) can be determined by placing pathogen inoculated plant material in water and measuring the increase in conductivity in the bathing solution over time (Baker et al., 1991; Mackey et al., 2002). This is now considered a standard method to quantify HR-PCD and a large proportion of the papers on the topic of plant pathogen defense include one or several such experiments. The pathogenic bacteria in this type of experiments are typically delivered into the plant leaf by pressure infiltration through the stomata using a needleless syringe. The leaves are either directly or after some time detached from the plant and discs are punched out with a cork borer. The discs are subsequently left to wash for some time, placed in water, and the increase in conductance of the bathing water measured over a period of hours or even days (Hibberd, Stall & Bassett, 1987). Hand infiltration of leaves is rather tedious work and does not allow multiple plant lines, or even replicates, to be inoculated simultaneously. For us it usually takes about 45 min from the first leaf is infiltrated to the last for an experiment consisting of five plant lines with six replicates (four leaf discs per replicate). If exact intervals for the time points of measurement are to be kept, it is required that conductivity is measured about every five minutes throughout the required time period. For a normal effector response this is about four to eight hours of constant measuring. In this context, we felt that there was room for improvement of the quantification method of HR-PCD.

In here, we present a quick and simple vacuum-based infiltration method of leaf material suitable for quantification of HR-PCD and other defense associated responses in plants. The method was used to test how cultivation conditions of the bacterial pathogen Pseudomonas syringae influence its ability to elicit HR-PCD in Arabidopsis. To further evaluate the applicability of this method, we investigated the role of bacterial titer on the outcome of HR-PCD. We found that lower bacterial inocula did not only result in decreased cell death and electrolyte leakage in plants as previously reported, but also resulted in a delay of the full defense reaction. The presented vacuum infiltration method is time-saving over syringe inoculation, shows higher reproducibility, and allows instantaneous infiltration of multiple plant lines. The method may also be used on material which has been pretreated with toxic or harmful reagents, abolishing the need for manual handling of and minimizing the exposure to the experimenter.

Materials and Methods

Plant material and growth condition

Arabidopsis thaliana wild type Columbia-0 (Col-0) and rpm1.3 (Grant et al., 1995) were sown in soil, stratified at 4 °C for 72 h and cultivated in growth chamber under short day conditions (8 h day/16 h night, 22 °C/18 °C) at 60% relative humidity. Seedlings were transplanted into individual pots two weeks after germination. Plants used in experiments were 5–6 weeks old.

Vacuum infiltration

Leaf discs were prepared using a cork borer (7 mm diameter) by punching them out from detached leaves against a piece of Styrofoam. Approximately 35 discs from 3 individual plants were made for each plant line/treatment and placed in 50 ml plastic centrifuge tubes. Thereafter discs were covered in 20 ml of the bacterial suspension and immediately placed in a SpeedVac vacuum concentrator (Savant, Thermo Electron Corporation, Waltham, MA, USA) with the rotor removed. It is important that the lids of the centrifuge tubes are loosely tightened to allow free exchange of air between the tube and the surrounding, while keeping the solution and the discs in the tube. To avoid aerosolized vapor containing bacteria in the laboratory environment, the vacuum concentrator was connected to a vapor trapping system. Pressure was decreased until the bacterial suspension started boiling at room temperature and kept as such for 10 s. Pressure was then quickly increased and samples brought back to atmospheric conditions. Note that the majority of the discs are infiltrated when the negative pressure is removed. We usually observe between 75 and 100% efficiency in infiltration, i.e., discs that lose their buoyancy. If the infiltration efficiency is low, the vacuum treatment may be repeated. There is no need for the addition of a surfactant to aid the infiltration, in fact, this will cause the bacterial suspension to foam and limit contact with the leaf discs. After infiltration, discs were poured into a tea strainer, rinsed in deionized water and transferred to Petri dishes containing water. Finally, fully infiltrated discs (sunken discs) without any visible damage from the coring were transferred using plastic bacterial inoculation loops to 6 well cell cultivation plates filled with 10 ml deionized water (four discs in each well in 6 replicates per treatment). Conductivity was measured by moving 5 ml of the bathing solution to a 15 ml centrifuge tube in which a conductivity meter (Orion, Thermo scientific, Waltham, MA, USA) was placed. The solution was thereafter returned to the plate well.

Pseudomonas culturing, cell staining and image acquisition

Pseudomonas syringae pv. tomato DC3000 expressing the effector protein AvrRpm1, AvrRpt2 or AvrRps4, or a strain carrying an empty vector (DC3000) (Bisgrove et al., 1994; Grant et al., 1995) were propagated on solid Pseudomonas agar F (King’s B medium, Biolife, Milano, Italy) containing kanamycin (50 mg l−1) and rifampicin (50 mg l−1) in a drawer in darkness at room temperature unless stated otherwise. P. syringae hrcC mutant (Rahme, Mindrinos & Panopoulos, 1991) were grown without kanamycin. Bacteria were re-plated 24 h prior to infiltration into plants. A loopfull of the overnight culture of bacteria was suspended in 30 mL of 10 mM MgCl2 and optical density at 600 nm was measured using a NanoDrop 2000c (Thermo Scientific, Waltham, MA, USA) in the cuvette mode. OD600 = 1.0 correspond to 3.55∗108 CFU ml−1 as determined by serial dilution and plating. Trypan blue staining of discs was adapted from (Koch & Slusarenko, 1990). Four discs were placed in a micro centrifuge tube, covered with staining solution (0.025% trypan blue in equal volumes phenol, glycerol, lactic acid and water) and kept at 95 °C in a heating block for 2.5 min. Samples were left in the staining solution for 30 min, then rinsed once in water and thereafter de-stained for two days in chloral hydrate:water (5:2 (w/v)) on an orbital shaker at 50 rpm. Stained discs were mounted in 60% (v/v) glycerol and images were acquired using a Zeiss Axioplan 2 Imaging microscope with a Plan-Neofluar 10x/0.30 objective connected to a Canon Powershot G6 camera. Images were subjected to Auto color filtering in Adobe Photoshop CS5 (Adobe Systems Incorporated, San Jose, CA, US).

Quantification of jasmonic acid and free and esterified OPDA

In tissue quantification of glycerolipid-bound 12-Oxo-phytodienoic acid (OPDA) was performed as previously described (Nilsson et al., 2012). For the quantification of free OPDA and JA released by tissue, 6 leaf discs were placed in a glass tube with 2 ml dH2O and kept on light agitation on an orbital shaker. Discs were removed at indicated times and samples were acidified with 50 µl 1.6 M HCl follow by phase separation through the addition of 2 ml ethyl acetate. The organic phase was collected and the original sample was re-extracted with 2 ml of ethyl acetate before drying under a stream of nitrogen gas. Samples were dissolved in 50 µl methanol, run on an Agilent 1290 Infinity LC coupled to an Agilent 6410 triple quadrupole mass selective detector and the amounts of JA and free OPDA were quantified as described (Pan, Welti & Wang, 2010), except that hexadeuterated JA (OlChemim Ltd, Olomouc, Czech Republic) was used as internal standard.

Quantification of phosphatidic acid

For labeling experiment, leaf discs were incubated with 33PO4 (New England Nuclear, Perkin Elmer) over night as described (Andersson et al., 2006). Discs were then transferred to 50 ml centrifuge tubes and vacuum infiltrated with P. syringae expressing AvrRpm1 at OD600 = 0.1 as described above. Mock treatment was performed by inoculating plants with 10 mM MgCl2. Lipids were extracted as previously described (Andersson et al., 2006). Samples were loaded on activated thin layer chromatography plates and developed using the solvent system chloroform:methanol:acetic acid:water (80:20:10:3.5), and subjected to autoradiography. Developed films were scanned and the intensity of the phosphatidic acid bands was determined using the western blot tool in ImageJ (http://rsb.info.nih.gov/ij/). Unlabeled PA was measured in a total lipid extract obtained from two leaf discs by LC-MS/MS. A total lipid extract was obtained as described (Andersson et al., 2006) and the lipids separated by reverse phase chromatography and detected by MS/MS on a triple quadrupole detector as described (Nilsson et al., 2014) using the neutral loss of m/z 115 (Li-Beisson et al., 2010; Schwudke et al., 2006) to detect PA species.

Statistical- and regression analysis

Data were handled in Excel (Microsoft, Redmond, WA, USA). Variances are expressed as standard deviation and significance was determined by Student’s t-test. Modeling of the electrolyte leakage from Col-0 was performed in SAS version 9.1 (SAS Institute, Cary, NC, USA) using the five parameter Weibull regression model with a Box–Cox transformation of the concentration (Eq. (1)) (Scholze et al., 2001). (1) effect=θmin+θmax−θmin×1−exp−expθ1+θ2×tθ3−1θ3,

where t represents time after vacuum infiltration, θmin is the lower asymptote, θmax the upper asymptote, and θ1−3 are additional parameters that describe the curve itself. Fixed parameters for the regression analysis are presented in Table S2.

Results and Discussion

Vacuum infiltration does not cause more wounding than syringe inoculation

The aim of this study was to develop a simple and quick protocol for scoring plants for effector-triggered defense responses. Vacuum infiltration has previously been used as a means to deliver bacteria into the leaf tissue of whole plants for the quantification of resistance and ETI (Coll et al., 2010; Yu, Katagiri & Ausubel, 1993). Vacuum has also been applied to infiltrate leaf discs with a bacterial suspension for determination of hypersensitivity and electrolyte loss (Brisset & Paulin, 1991; Dellagi et al., 1998; Venisse, Gullner & Brisset, 2001). However, the experimental procedure described in these reports is rather cumbersome and the throughput of the systems is limited. We decided to examine the use of vacuum to deliver a bacterial pathogen into leaf discs of several plant lines and replicates simultaneously using standard lab equipment (Fig. 1). A SpeedVac vacuum concentrator connected to a vapor trap was used to create the needed vacuum for the infiltration. Pressure was reduced until the bacterial suspension with the submerged leaf discs reached the boiling point at room temperature (approximately 80–100 kPa below ambient atmospheric pressure). The majority of the discs were infiltrated when they were quickly brought back to normal atmospheric pressure. An easy to follow step-by-step protocol of the procedure is available as a Supplemental Information 1.

Figure 1 Schematic representation of the vacuum infiltration method.

Leaf discs are punched out and placed in plastic centrifuge tubes. Bacterial suspension is added to the discs and vacuum is applied. The discs are then poured into a tea strainer, washed with deionized water and transferred to Petri dishes filled with water. Fully infiltrated discs, distinguishable by their loss of buoyancy, are then transferred to 6 well cell cultivation plates and the conductivity of the bathing solution is monitored.

In order to evaluate the use of vacuum for pathogen delivery into leaf discs, we first set out to determine and compare the wounding effect of syringe versus vacuum inoculation. To this end we measured the formation of jasmonic acid (JA) and both free and lipid-bound forms of its precursor 12-oxo-phytodienoic acid (OPDA), as these jasmonates are highly induced by mechanical wounding in Arabidopsis (Buseman et al., 2006; Glauser et al., 2009; Kourtchenko et al., 2007; Nilsson et al., 2012; Stelmach et al., 2001). Discs made from leaves previously syringe infiltrated with 10 mM MgCl2 were compared to discs that had been submerged in MgCl2 and subjected to vacuum infiltration. As a positive control, leaf discs were frozen in liquid nitrogen and left to thaw at room temperature since this treatment activates a strong wounding response in terms of accumulation of lipid-bound OPDA (Nilsson et al., 2012). The amounts of free OPDA, lipid-bound OPDA and JA were quantified before and after infiltration (Figs. 2A–2C). Neither of the two infiltration procedures caused any detectable accumulation of lipid bound OPDA, whereas freeze-thawing, as expected, caused a large increase in bound OPDA. Furthermore, only minute amounts of free OPDA were released by vacuum and syringe inoculated plant tissue while freeze-thawed samples released approximately 30 nmol per gram tissue after 4 h. JA levels increased following both vacuum and syringe treatment to similar concentrations. As previously reported, freeze-thaw wounding did not trigger the accumulation of JA (Glauser et al., 2009). Discs were also stained for cell death using trypan blue 0.5 and 4 h post inoculation (Fig. 2D). No cell death was apparent 0.5 h after treatments. At 4 hpi, cells at the edges of the discs that had been wounded by the borer during tissue preparation stained positive. It seems reasonable that these cells are the source of the low amounts of JA and OPDA detected in syringe- as well as vacuum inoculated samples. The presented experiments demonstrate that vacuum infiltration does not trigger a significant wounding response in Arabidopsis in terms of production of jasmonates. At the very least, the vacuum method does not cause a stronger wounding response than syringe infiltration. It should be noted that avoiding tissue damage during syringe infiltration takes a bit of practice, whereas vacuum infiltration in this respect can be considered as more or less foolproof and can be successfully performed by first time users.

Figure 2 Wounding response after syringe and vacuum infiltration.

Leaf discs from 5 weeks old Arabidopsis plants were either syringe or vacuum infiltrated with 10 mM MgCl2. Control samples were frozen in liquid nitrogen and left to thaw at room temperature. Leaf discs were analyzed for glycerolipid-bound OPDA at indicated times (A). Free OPDA and JA released by plants are shown in (B) and (C), respectively. (D) Shows dead cells in leaf disc after trypan blue staining. Error bars represent standard deviation from replicate samples.

Cultivation conditions influences Pseudomonas ability to elicit HR-PCD

Pathogenic strains of the Gram-negative bacterium Pseudomonas syringae expressing the effector AvrRpm1 cause a strong HR in Arabidopsis mediated through the R-protein RPM1 (Debener et al., 1991). Conversely, the RPM1 protein null mutant rpm1.3 is unable to recognize the effector and thus incapable of mounting a significant HR (Grant et al., 1995). This is an extensively studied host-pathogen model system (Katagiri, Thilmony & He, 2002) and was chosen to evaluate vacuum infiltration as a method to study ETI.

We wanted to test how the cultivation conditions of the bacteria used for the vacuum infiltration affect the defense reaction of the plant; this in order to standardize the procedure. Pseudomonas species are typically cultivated in King’s Medium B based on the formulation of King and colleagues (1954). The exact nutritional composition of Pseudomonas media does however differ between manufacturers. Using vacuum infiltration, we tested how different media recommended for Pseudomonas cultivation influenced the ability of P. syringae pv. tomato DC3000 expressing AvrRpm1 to elicit HR-PCD in Arabidopsis. Bacteria grown over night at room temperature on three different commercially available solid media (Table S1) were suspended in 10 mM MgCl2 (OD600 = 0.1, 3.55∗107 CFU ml−1) and infiltrated into leaf discs of wild-type Col-0 plants. Discs were washed in deionized water for 10 min, placed in 6 well culture plates filled with water and the conductivity of the bathing solution measured hourly for a total of eight hours (Fig. 3A). Bacteria cultivated on Pseudomonas agar F (Biolife, Milano, Italy) elicited the strongest plant defense response in terms of electrolyte leakage and was selected for use in further experiments. Notably, bacteria cultivated on Pseudomonas agar base (Biolife, Italy) elicited only approximately half the ion leakage and initiation of the response reaction appeared to be delayed.

Figure 3 Media and temperature-dependent ability of P. syringae to elicit HR.

P. syringae DC3000 expressing the effector AvrRpm1 cultivated on three types of solid media (A) or at two different temperatures (B) were vacuum infiltrated into wild-type Col-0 leaf discs (OD600 = 0.1). Culture media used in B was Pseudomonas agar F (King’s B medium, Biolife, Milano, Italy). Mean and standard deviation where n = 6 (A) and n = 6 (B) are shown. Experiments were replicated with similar results.

It has been reported that P. syringae pv. glycinea synthesizes substantially higher levels of the phytotoxin coronatine when grown at 18 °C as compared to growth at 28 °C (Palmer & Bender, 1993). Likewise, the AvrPto effector from P. syringae pv. tomato is secreted when the bacteria are grown at 20 °C but not at 30 °C (Van Dijk et al., 1999). On the other hand, pre-inoculation temperature did not influence P. syringae strains’ ability to cause visual HR symptoms on tobacco (Budde & Ullrich, 2000). Next, we tested whether the growth temperature of P. syringae DC3000 avrRpm1 had any effect on the outcome of the HR on Arabidopsis (Fig. 3B). The HR-PCD was found to be significantly delayed when the bacteria had been cultivated at 28 °C compared to when grown at 21 °C (room temperature). The underlying mechanism to this thermosensitivity may be that the bacteria produce more toxins and have higher secretion of effectors at lower temperature as discussed above. Another possibility is that bacteria grown at the higher temperature will reach stationary phase at an earlier time point and that this influences their ability to induce HR. Taken together, these results demonstrate that differences in the cultivation of the pathogen can have profound impact on the outcome of the HR. This is something experimenters should consider and standardize as far as possible. Even rather subtle changes such as changes in batches of media or unstable growth temperature might affect the reproducibility substantially.

Vacuum infiltration shows high reproducibility

A key feature of any method is that it shows robustness and that reproducibility is high. To evaluate the vacuum method in this respect, data retrieved from 13 independent experiments where plants had been inoculated with P. syringae DC3000 avrRpm1 at OD600 = 0.1 (3.55∗107 CFU ml−1) using syringe or vacuum were compared. The deviation from the average in each experiment was determined 6 h after inoculation and represented in a box plot (Fig. 4). Both the maximum deviation from the median and the distribution around the median were lower when plant material had been vacuum infiltrated with the pathogen. Thus, at least in our hands, reproducibility is higher when vacuum is used as compared to syringe inoculation.

Figure 4 Reproducibility in vacuum and syringe infiltration experiments.

Data retrieved from 13 independent experiments where plants were inoculated with P. syringae DC3000 AvrRpm1 at OD600 = 0.1 using syringe or vacuum are presented in the chart. Dots show the deviation from average 6 h after infiltration of the pathogen in individual experiments. Boxes show the deviation from average in the 10–90% percentiles and the lines within boxes indicate median values.

Vacuum infiltration can be used to measure HR elicited by AvrRpt2 and AvrRps4

The AvrRpm1 effector induces a relatively fast HR in wild-type Col-0 plants (Debener et al., 1991; Grant et al., 2000). To extend the applicability of the vacuum method, electrolyte leakage was monitored after inoculation of two P. syringae DC3000 strains carrying effectors that are considerably slower in promoting HR than AvrRpm1, viz., AvrRpt2 and AvrRps4 (OD600 = 0.1, 3.55∗107 CFU ml−1) (Fig. 5). Discs infiltrated with P. syringae carrying the empty vector DC3000, MgCl2 or uninfiltrated samples were also included in the experiment. At the end of the measurements (24 h), no significant difference in conductivity was observed between untreated samples and samples mock inoculated with MgCl2. Again demonstrating that the vacuum procedure does not cause more cell death and loss of electrolytes than the tissue preparation in itself. A clear induction of the HR was noted in samples infiltrated with bacteria expressing AvrRpt2 after eight hours. The same reaction was observed slightly later in AvrRps4-treated samples. At 24 h after infiltration, there was no significant difference in samples inoculated with virulent DC3000 and those inoculated with AvrRpt2 or AvrRps4. P. syringae pv. tomato DC3000 carries approximately 300 confirmed and putative virulence genes (Buell et al., 2003). P. syringae lacking the type III secretion system (hrcC mutant) did only provoke a very small release of electrolytes from leaf discs over 24 h (Fig. S1). The relatively high increase in conductivity in samples treated with DC3000 empty vector may thus be ascribed to secreted toxins and/or effectors other than AvrRpm1, AvrRps2 and AvrRps4. The rapid growth of virulent DC3000 will also cause necrotic host cell death and subsequently electrolyte loss (Grant et al., 1995). The results show that the vacuum infiltration method works well also with effectors which induce considerably slower HR responses than AvrRpm1.

Figure 5 Electrolyte leakage in leaf discs treated with P. syringae carrying different effectors.

Leaf discs from wild-type Col-0 plants were vacuum infiltrated with P. syringae DC3000 carrying an empty vector (DC3000) or the effector AvrRpt2 or AvrRps4. Discs in control samples were infiltrated with 10 mM MgCl2 or left untreated. Mean and standard deviation where n = 6 are shown. The experiment was replicated with similar results.

The influence of bacterial titer on the outcome of the HR

The concentration of pathogenic bacteria used for electrolyte leakage experiments is typically several orders higher than the inocula that plants are exposed to in nature (Hirano & Upper, 2000). To be able to mimic natural plant-microbe interactions more closely in the lab, we sought to investigate the influence of the bacterial titer on the hypersensitive cell death response. Already some forty years ago, Turner & Novacky (1974) recognized that there exists a strong correlation between the inoculum of incompatible bacteria and the number of dead plant cells after the HR. However, the dynamic changes in cell death that follow effector-triggered HR at different bacterial titers have to the best of our knowledge not been thoroughly investigated. To address this, five concentrations of P. syringae expressing AvrRpm1, from the same overnight culture, were simultaneously vacuum infiltrated into leaf discs of wild-type Col-0 and rpm1.3 plants (Figs. 6A and 6B). In addition, discs were trypan blue stained for visualization of cell death (Fig. 6C). As expected, initiation of the HR, and consequently a dramatic increase in conductivity, was observed 2–3 h after infiltration with the two highest bacterial titers (OD600 = 0.05 and 0.1, corresponding to 1.78 and 3.55∗107 CFU ml−1, respectively). The rapid conductivity change was accompanied by a massive cell death where almost all of the mesophyll cells stained dark with trypan blue three hours after inoculation. A more moderate increase in electrolyte leakage was observed in Col-0 plants treated with lower bacterial titers (OD600 = 0.001–0.01, 3.55∗105–3.55∗106 CFU ml−1). No statistical difference in the maximum electrolyte leakage and the appearance of cell death from trypan blue staining were observed when wild-type was inoculated with bacterial titers as high or higher than OD600 = 0.05. In this context, it is tempting to speculate that mutants hyperactive in HR or mutants that display kinetic differences in HR-PCD associated release of electrolytes may have been missed or overlooked in studies where high bacterial inoculum has been used. This since the number of bacteria necessary for activating HR-PCD is significantly lower than what is commonly used. Electrolyte release, as measured by conductivity, remained low at all concentrations throughout the experiment in rpm1.3 samples, although a small dose-dependent increase in electrolyte leakage was noticed. This increase in electrolytes is likely the result of activation of the plant RPS2 protein and other R proteins by the AvrRpm1 effector (Kim et al., 2009) and responses evoked by bacterial PAMPs and other effectors delivered by the pathogen.

When tested, it took about 45 min for one person to prepare 5 plant lines with 6 replicates using syringe infiltration. Preparation of the same number of lines and replicates for the vacuum infiltration assay was found to be slightly faster. Although the two methods required approximately the same amount of time for experiment start-up, they differ in that the pathogen treatment is instantaneous in all samples using vacuum infiltration. When plants were syringe infiltrated, approximately 40 min passed from the start to the finishing of the inoculation procedure. With such a long time span between the samples, it is necessary to make conductivity measurements approximately every five minutes throughout the experiment. Using vacuum infiltration, on the other hand, all samples are simultaneously inoculated and only about ten minutes of measurements every hour is required. We conclude that the vacuum infiltration method allows synchronized treatment in multiple plant lines and replicates and shows higher reproducibility than pressure infiltration, thus making it suitable for detection of subtle differences in HR-mediated electrolyte leakage between samples. Furthermore, the vacuum method requires less training and is not as time consuming as syringe infiltration.

Figure 6 Influence of bacterial titer on the outcome of HR.

Leaf discs from wild-type Col-0 (A) or rpm1.3 (B) Arabidopsis were simultaneously vacuum infiltrated with five concentrations (OD600 = 0.001–0.1) of P. syringae DC3000 expressing the effector AvrRpm1. Mean and standard deviation where n = 6 are shown. At indicated times, discs were stained with trypan blue for visualization of induced cell death (C). All images are shown at same scale. Scale bar indicates 500 µm.

Modeling the kinetics of HR-induced cell death

We next sought to study the temporal aspect of the HR and how it relates to bacterial titer in closer detail. The data set presented in Fig. 6A with the AvrRpm1 effector was used for this analysis. Since a RPM1 independent release of electrolytes could be observed during the first hour of conductivity measurements (Figs. 6A and 6B), only the data points from 1 to 8 h after infiltration were considered. A five parameter Weibull regression model with a Box–Cox transformation of the time was fitted to the Col-0 electrolyte leakage data (see ‘Material and Methods’ for details) (Figs. 7A–7E). The derivative of the functions was plotted for visualization of the strength and the timing of the HR (Fig. 7F). Here, peak values represent the maximum release rates of electrolytes from the plant tissue at the different bacterial titers. Figure 7G shows the second normalized derivate of the functions where the maximum release rate is apparent as intersect of the curve and the X-axis. The maximum electrolyte release rate was found to occur over two hours later in plant material inoculated with the lowest concentrations of bacteria compared to the highest. There was also a tendency for a delayed onset of the HR seen as a shift in maxima in the lower titer inoculations. These results were confirmed in two additional independent experiments where plant material was inoculated with P. syringae expressing AvrRpm1 at three different concentrations (OD600 = 0.1, 0.01 and 0.001) and conductivity measured every 30 min for a total of eight hours (Fig. S2). At least two not mutually exclusively explanations for this dose-dependent response may be put forward. In the first scenario, a certain number of bacteria must infect a plant cell before a threshold is reached and the plant initiates the cell death program. Thus, the amplitude of the electrolyte leakage will be lower and the initiation of the HR reaction will be slower in samples treated with low concentrations of bacteria. The second possibility is that the bacterially induced HR is initiated in certain cells and then spreads to neighboring cells. Such spread of PCD from primary infected cells to adjacent cells layers has been suggested previously (Andersson et al., 2015; Coll et al., 2010; Turner & Novacky, 1974). This would then appear as a shift towards a later onset of the defense reaction, and also a delay in maximal release rate of electrolyte at lower bacterial inocula. The appearance of the trypan blue stains for the two lowest bacterial titers lend some support to the latter explanation (Fig. 6C).

Figure 7 Modeling of the hypersensitive response reveals bacterial dose dependence.

A regression analysis was performed on the electrolyte leakage data from Col-0 presented in Fig. 6A. (A)–(E) Shows functions fitted to the data from the five different titers of P. syringae DC3000 AvrRpm1 where (A) shows OD600 = 0.1, (B) OD600 = 0.05, (C) OD600 = 0.01, (D) OD600 = 0.005, and (E) OD600 = 0.001. The first derivative of the functions displayed in (A)–(E), which shows the rate of change in electrolyte leakage over time, is shown in (F). The second normalized derivative of the functions displayed in (A)–(E) is presented in (G). Dots in (A)–(E) are actual measured values and dashed lines indicate 95% confidence intervals. Modeling was performed on two different data sets with similar results.

Vacuum infiltration as a tool to study PLD activity

Phosphatidic acid (PA) is a key signaling lipid involved in plant responses to both abiotic and biotic stresses (Arisz, Testerink & Munnik, 2009; Li, Hong & Wang, 2009). We have previously reported that PA is formed quickly in tissue after elicitation with AvrRpm1 and that PA by itself can induce HR-like cell death in Arabidopsis (Andersson et al., 2006). In the study we used a transgenic system with in planta expression of the AvrRpm1 effector under the control of a dexamethasone inducible promoter (Mackey et al., 2003; Mackey et al., 2002). Prior to induction of the effector gene, lipids in leaf discs had been labeled with 33PO4 or 14C-acetate. This experimental setup, with ectopic expression of AvrRpm1, eliminated the need for manual handling of radioactive leaf material, something which would have been necessary if leaves were pressure infiltrated with a bacterial suspension after radiolabeling. Additionally, we have repeatedly failed to obtain conclusive results on phospholipase activation after syringe infiltration. We thus tested if the herein described vacuum infiltration procedure was suitable for determining if an increase in PA, as apparent using the transgenic system, could be observed after treatment with P. syringae expressing AvrRpm1. Leaf discs labeled over night with 33PO4 were vacuum infiltrated with P. syringae DC3000 avrRpm1 (OD600 = 0.1, 3.55∗107 CFU ml−1) or mock treated with 10 mM MgCl2 and placed in deionized water. Lipids were extracted 0.5 and 4 h after infiltration and subsequently subjected to thin layer chromatography and autoradiography. Films were scanned and the intensity of the radiolabeled PA was determined (Fig. 8A). A five-fold, significant (Student’s t-test, p < 0.05), increase in PA radiolabel could be observed after 4 h in discs treated with the bacterium. PA levels in mock samples remained low over the four hours of the experiment. To support this, leaf material from Col-0 and rpm1.3 plants was vacuum infiltrated with P. syringae DC3000 avrRpm1 and analyzed for PA content with LC-MS/MS (Fig. 8B). A similar significant increase in PA as seen after radiolabeling was noted. The results show that radiolabeled PA accumulates quickly in plants also after elicitation with avirulent bacteria expressing AvrRpm1, and that vacuum infiltration provides a safe and reliable way of inoculating plants when syringe infiltration is not an option. The presented method allows a simple route for pre-treatment of leaf discs with for instance inhibitors or radiolabeled chemicals.

Figure 8 Phosphatidic acid accumulates in plants in response to P. syringae expressing AvrRpm1.

Arabidopsis leaf tissue was analyzed for the lipid second messenger phosphatidic acid after treatment with the avirulent P. syringae DC3000 AvrRpm1 at OD600 = 0.1. In (A), Arabidopsis leaf discs were overnight labeled with 33PO4 and vacuum inoculated with P. syringae or mock treated with 10 mM MgCl2. Phospholipids were extracted, separated on thin layer chromatography and exposed on X-ray films. Phosphatidic acid was quantified from developed films. In (B), leaf discs from Col-0 and the RPM1 loss-of-function mutant rpm1.3 were vacuum infiltrated with the bacteria and extracted lipids were analyzed by LC-MS. Samples were prepared in triplicates and error bars show standard deviation. Asterisk denotes p < 0.05, statistically significant difference between mock/rpm1.3 and pathogen treated wild-type at 4 hpi as determined by Student’s t-test.

Conclusions

The term hypersensitiveness was introduced by Stakman (1915) a century ago to describe the rapid cell death in cereal plants that followed inoculation with incompatible isolates of rust fungi and the last decade has seen an explosive increase in our understanding of the molecular details of the process (Mur et al., 2008). Various methods have been developed over the years to quantify the extent of the HR-associated PCD in plants induced by pathogens. A standard procedure to determine the HR is to measure electrolyte leakage after pressure infiltration with a suspension of avirulent bacteria. In this report we present a quick and effective protocol for the inoculation of bacteria by vacuum infiltration. Using vacuum infiltration, we show that the kinetics of the HR-PCD correlates to the density of effector-expressing P. syringae and we propose that this may be an important factor to consider when screening for mutants with perhaps weak or conditional phenotypes. The results also show that the status of the pathogen is important, as demonstrated by the effect of different media compositions and temperatures during pre-culture. However, we acknowledge the existence of additional factors, beyond our control, that affect both the kinetics and final amplitude of the conductivity curve.

We further demonstrate that the method is applicable to experiments when manual handling of the leaf material is preferentially avoided. The vacuum infiltration protocol presented here can be combined with several types of analytical methods and we have successfully used it in our lab to monitor early changes in metabolite content, gene expression and protein abundance after pathogen and chemical elicitation. The method is particularly useful when large quantities of inoculated leaf material are needed, for example when quantifying low abundant metabolites. Finally, we believe the described method will be an excellent tool to screen large populations of Arabidopsis mutants for aberrant defense response phenotypes.

Supplemental Information

Figure S1 Electrolyte leakage from leaf discs vacuum inoculated with virulent P. syringae and hrcC mutant

Wild-type Col-0 Arabidopsis leaf discs were vacuum infiltrated with virulent P. syringae DC3000 or the hrcC mutant lacking type III secretion system at OD600 = 0.1. Mean and standard deviation where n = 6 are shown. The experiment was replicated with similar results.

Click here for additional data file.

Figure S2 Modeling of the hypersensitive response induced by P. syringae expressing AvrRpm1 at three different concentrations reveals bacterial dose dependence

A regression analysis was performed on electrolyte leakage data from two independent experiments (A and B, respectively) where leaf discs from Col-0 plants had been infiltrated with P. syringae expressing the effector AvrRpm1 at three different concentrations.

Click here for additional data file.

Data S1 Raw data

Click here for additional data file.

Supplemental Information 1 Step by step protocol

The document describes a step by step protocol for the method described in the article.

Click here for additional data file.

Table S1 Pseudomonas growth media composition

Nutritional composition of three different media for cultivation of Pseudomonas sp. Numbers inducate g/L media as reported by the manufacturers.

Click here for additional data file.

Table S2 Parameter estimates from regression analysis

Given in the table are the estimated parameters used for modeling electrolyte leakage from Col-0 plants inoculated with P. syringae expressing the AvrRpm1 effector.

Click here for additional data file.

We thank Professor Murray Grant for Pseudomonas strains.

Additional Information and Declarations

Competing Interests

Author Contributions

Data Availability

Thomas Backhaus is an Academic Editor for PeerJ.

Oskar N. Johansson and Anders K. Nilsson conceived and designed the experiments, performed the experiments, analyzed the data, wrote the paper, prepared figures and/or tables, reviewed drafts of the paper.

Mikael B. Gustavsson conceived and designed the experiments, performed the experiments, analyzed the data, wrote the paper, reviewed drafts of the paper.

Thomas Backhaus conceived and designed the experiments, performed the experiments, analyzed the data, contributed reagents/materials/analysis tools, wrote the paper, reviewed drafts of the paper.

Mats X. Andersson analyzed the data, contributed reagents/materials/analysis tools, wrote the paper, reviewed drafts of the paper.

Mats Ellerström conceived and designed the experiments, analyzed the data, contributed reagents/materials/analysis tools, wrote the paper, reviewed drafts of the paper.

The following information was supplied regarding data availability:

The research in this article did not generate any raw data apart from that in the included Data S1.

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
