# Peer review of "A quick and robust method for quantification of the hypersensitive response in plants"

_PeerJ, doi:10.7717/peerj.1469_

## Round 0.1 · original submission · Minor Revisions

The paper has been appreciated by the reviewers and positive comments have been expressed. However, the reviewers have proposed some modifications you should address to improve the quality of the paper.

Reviewer 1 ·

Basic reporting

The article is well written and easy to follow and adheres to PeerJ policies to the best of my judgement. The introduction gives a good background of the methods used to this end in monitoring immunity-related PCD - HR, and refers to existing literature adequately. Figures are clear and relevant, and overall the study represents a coherent and wide investigation of the suitability of a particular method development.

Experimental design

The research question is clearly defined as to whether vacuum infiltration can substitute syringe infiltration when studying HR elicited by P.s. DC3000 secreting avirulence proteins avrRpm1, Rpt2, Rps4. I appreciate in depth investigation of how the used bacterial density affects HR, and contribution of media and cultivation as we many times see variability in the strength of HR between experiment, evident also from the raw data comparing vacuum with syringe infiltration between 13 independent experiments. Except for the deviation graph that compares vacuum with syringe infiltration, I find that the experiment design did not compare syringe and vacuum infiltration. I suggest that the raw data of figure 4 is used to show variability in the strength of HR between syringe and vacuum infiltration which appears to be quite in the same range, but also use the 13 experiments to illustrate the high variability between experiments despite the effort of standardizing.

Validity of the findings

Overall I feel confident about the validity of the presented data. Because this is a methods paper, I think reproducibility between experiments need to be emphasized, as I also interpret the authors wish. I assume that the experiments have been repeated, and I propose that the raw data from at least one repetition is given in the raw data sheet as done with figure 6 + supplement 2. Additionally, I would have appreciated that avrRpt2 and Rps4 triggered ion leakage would have been compared vacuum vs. syringe, but this is not very essential as the leakage curves look "syringe-like".

Additional comments

I think you have addressed a critical problem how to reduce variability by standardizing the ion leakage HR assay. An important issue that you should discuss is how variable the amplitude of HR is also in your experiments, assuming that I interpret figure 4 raw data correctly, and figure 6/raw data S2.This reflects on the true variability we face with this assay and should be discussed in the conclusions.

Reviewer 2 ·

Basic reporting

No comment

Experimental design

OK

Validity of the findings

See general comments

Additional comments

The authors propose an improved method to quantify the plant hypersensitive response that will not only increase the throughput but also reduce time limitations and supress the labour intensive step. Cell death following pathogen elicitation is evaluated through electrolyte leakage, measured as an increase in conductivity. As such study of this kind is very much pertinent and of interest to the plant/micro-organisms research community.

However the authors suggest that (lines 23-26, 43-44, 61-62, …) PCD is mandatorily activated during incompatible interaction. This point, namely the importance of cell death to resistance, remains to be resolved even today. Furthermore significant electrolyte leakage also occurs following the production of necrotic lesions during a compatible interaction. It also appears that the dynamics of the signal i.e, the changes in conductivity do not allow to discriminate between PCD induced by different effectors (see fig. 2 & 5 ), and necrotic lesions. So the method appears to measure cell death whatever the underlying mechanism.
I think this point so be further discussed in the MS before publication.

Minors
Line 105-108. The use of a Pseudomonas syringae with an empty vector should be mentioned in the MM section too.
Line 170. A bacterial suspension: more appropriate than bacterial solution, see line 176.
Line 267. Pseudomonas syringae DC300 expressing ? , carrying ? empty vector
Line 268. Uninfiltrated instead of unifiltrated
Lines 306-310. Not clear to me
Line 322- 326. Here again I agree that the method is “suitable for detection of subtle differences in electrolyte leakage “but it is by no means that these changes in electrolyte leakage are associated with HR in all cases.

---

## Round 0.2 · accepted · Accept

The authors have provided satisfying responses to the comments sent by the reviewers and therefore this paper is considered as acceptable for publication.

Reviewer 1 ·

Basic reporting

OK

Experimental design

OK

Validity of the findings

As before

Additional comments

I am satisfied with the authors response to my comments

Reviewer 2 ·

Basic reporting

no comment

Experimental design

no comment

Validity of the findings

no comment

---

## Author Rebuttal · Round 0.2

Gothenburg, October 19, 2015

Dear Editor,

Please find enclosed a revised version of the above referenced manuscript. We are grateful to the two anonymous reviewers for their constructive criticism; their input has enabled us to resubmit an improved manuscript. We have made revisions to the manuscript according to the suggestions from reviewers as marked in yellow in the document.

Our detailed response (Black) to the reviewers' comments (Red) follows below.

**Reviewer 1**

Except for the deviation graph (figure 4, authors' comment) that compares vacuum with syringe infiltration, I find that the experiment design did not compare syringe and vacuum infiltration. I suggest that the raw data of figure 4 is used to show variability in the strength of HR between syringe and vacuum infiltration which appears to be quite in the same range, but also use the 13 experiments to illustrate the high variability between experiments despite the effort of standardizing.

We did compare also the amount of time consumed by setup, amount of damage (release of OPDA, JA, TB stain). However, while there was a notable (10 min) difference in the time for setup, the real improvement is the simultaneous measurements that can be done as all material is infiltrated at the same time.

Figure 4 does not show variability in the strength of the HR, only deviation from the average value at the final measurement. There are some differences in the number of leaf discs and water volume used in the 13 experiments that figure 4 is based on. This is marked with an asterisk in the raw data file. We could therefore not compare the absolute values, i.e. the strength of the HR, between experiments. However, we show that regardless of number of discs and water volume, the vacuum method seems to be slightly more consistent compared to syringe infiltration.

I think reproducibility between experiments needs to be emphasized, as I also interpret the authors wish. I assume that the experiments have been repeated, and I propose that the raw data from at least one repetition is given in the raw data sheet as done with figure 6 + supplement 2

We have added raw data from independent experiments using the vacuum infiltration method for ion leakage assays (Figure 3A ,3B, 5). Fig S2 also shows two independent repetitions of the experiment shown in figure 6.

An important issue that you should discuss is how variable the amplitude of HR is also in your experiments, assuming that I interpret figure 4 raw data correctly, and figure 6/raw data S2.This reflects on the true variability we face with this assay and should be discussed in the conclusions.

We fully agree, and we acknowledge the existence of additional parameters still beyond our control that effect the amplitude. We have now added this to the discussion.

For variation in HR amplitude in the raw data for figure 4, please see our response above.

**Reviewer 2**

However the authors suggest that (lines 23-26, 43-44, 61-62, …) PCD is mandatorily activated during incompatible interaction. This point, namely the importance of cell death to resistance, remains to be resolved even today.

We agree that incompatible interactions are not necessarily defined by PCD and we have added a clarification of this in the manuscript. We have also clarified that it is the cell death associated with the HR that we are refereeing to and not the full HR.

Cell death *per se* do not appear to always be important for resistance against *Pseudomonas* carrying effectors, as found for example in the *defense no death* mutants (*dnd1* (Clough et al. 2000; Yu et al. 1998); *dnd2* (Jurkowski et al. 2004)) and the *penetration* (*pen*) mutants (Johansson et al. 2014).

Thus, the HR as a whole effectively stops many pathogens, regardless of the presence or apparent extent of the programmed cell death.

Furthermore significant electrolyte leakage also occurs following the production of necrotic lesions during a compatible interaction. It also appears that the dynamics of the signal i.e, the changes in conductivity do not allow to discriminate between PCD induced by different effectors (see fig. 2 & 5 ), and necrotic lesions. So the method appears to measure cell death whatever the underlying mechanism.

And minor comment

- Line 322- 326. Here again I agree that the method is "suitable for detection of subtle differences in electrolyte leakage "but it is by no means that these changes in electrolyte leakage are associated with HR in all cases.

It is true that one cannot discriminate between electrolytes released during PCD or any other type of loss of cellular integrity (for example, boiling leaf discs also releases electrolytes). Plants exposed to high light and heat during the HR release electrolytes in a linear fashion over time, suggestive of damage rather than effector-R associated PCD release. In the case of Effector-R interaction, the kinetics of electrolyte release are fundamentally different from the release caused by abiotic damage or necrotic lesions. Conductivity as a measure of HR related cell death has been used in the community for decades, this paper however, uses vacuum for delivery of the pathogen directly and simultaneously into plant tissue.

*Pseudomonas syringae* is a hemi-biotrophic pathogen and is able to proliferate if undetected by fast effector-R interactions. It has been shown that injected, undetected effectors such as AvrRpm1 induces production of endogenous sugar transporters in the plant cell and many effectors have been shown to actually prevent cell death measures taken by the plant cell (Jamir et al. 2004).

Though, it is important to use proper controls. For instance, the Arabidopsis mutant *rpm1-3* lacks the main R-protein used to detect disturbances by the effector AvrRpm1 (Fig. 6B). The *rpm1-3* mutant does

not initiate programmed cell death over the same period of time compared to wild type, and does not release electrolytes with the same kinetics, or amplitude. While it is true that some of the released ions can be due to other effectors carried by *Pseudomonas*, the notion that plants exposed to the *Pseudomonas* empty vector strain DC3000 only releases a fraction of the electrolytes compared to plants exposed to effector carrying strains (effectors AvrRpm1, AvrRpt2 or AvrRps4, see Fig. 5), supports that what we measure is indeed an active defense reaction from the plant.

- Lines 306-310. Not clear to me

We have tried to clarify this in text.

On lines 306-308 we suggest that there could potentially exist mutants that, for example, display increased or faster HR related PCD. As the standard bacterial inoculum is $OD_{600}$ = 0.1, there is a risk that these mutants, displaying altered kinetics and amplitude of the HR, would have been overlooked. Thus, we suggest that a range of different $OD_{600}$s could be used to screen for such mutants.

At lines 308-310 we suggest that when a high density ($OD_{600}$ = 0.1) of DC3000:AvrRpm1 is infiltrated into the *rpm1-3* mutant, there is often a small but significant increase of electrolyte leakage, compared to lower inoculum. This could be attributed to RPS2 activation by AvrRpm1 (Kim et al. 2009) or other R-proteins sensing AvrRpm1 action.

**The following minor comments have now been addressed as suggested:**

- Line 105-108. The use of a Pseudomonas syringae with an empty vector should be mentioned in the MM section too.
- Line 170. A bacterial suspension: more appropriate than bacterial solution, see line 176.
- Line 267. Pseudomonas syringae DC300 expressing ? , carrying ? empty vector
- Line 268. Uninfiltrated instead of unifiltrated

**Referenced literature**

Clough SJ, Fengler KA, Yu IC, Lippok B, Smith RK, and Bent AF. 2000. The Arabidopsis dnd1 "defense, no death" gene encodes a mutated cyclic nucleotide-gated ion channel. *Proc Natl Acad Sci U S A* 97:9323-9328.

Jamir Y, Guo M, Oh HS, Petnicki-Ocwieja T, Chen S, Tang X, Dickman MB, Collmer A, and Alfano JR. 2004. Identification of Pseudomonas syringae type III effectors that can suppress programmed cell death in plants and yeast. *Plant J* 37:554-565.

Johansson ON, Fantozzi E, Fahlberg P, Nilsson AK, Buhot N, Tor M, and Andersson MX. 2014. Role of the penetration-resistance genes PEN1, PEN2 and PEN3 in the hypersensitive response and race-specific resistance in Arabidopsis thaliana. *Plant J* 79:466-476.

Jurkowski GI, Smith RK, Yu IC, Ham JH, Sharma SB, Klessig DF, Fengler KA, and Bent AF. 2004. Arabidopsis DND2, a second cyclic nucleotide-gated ion channel gene for which mutation causes the "defense, no death" phenotype. *Molecular Plant-Microbe Interactions* 17:511-520.

Kim MG, Geng X, Lee SY, and Mackey D. 2009. The Pseudomonas syringae type III effector AvrRpm1 induces significant defenses by activating the Arabidopsis nucleotide-binding leucine-rich repeat protein RPS2. *Plant J* 57:645-653.

Yu IC, Parker J, and Bent AF. 1998. Gene-for-gene disease resistance without the hypersensitive response in Arabidopsis dnd1 mutant. *Proc Natl Acad Sci U S A* 95:7819-7824.